# Reduction in Xenogeneic Epitopes on Porcine Endothelial Cells by Periodate Oxidation

**DOI:** 10.3390/biomedicines12071470

**Published:** 2024-07-03

**Authors:** Jonas Thom, Nathalie Roters, Slavica Schuemann, Birgit Andrée, Falk F. R. Buettner, Andres Hilfiker, Tobias Goecke, Robert Ramm

**Affiliations:** 1Leibniz Research Laboratories for Biotechnology and Artificial Organs (LEBAO), Hannover Medical School, 30625 Hannover, Germany; 2Lower Saxony Centre for Biomedical Engineering, Implant Research and Development, 30625 Hannover, Germanygoecke.tobias@mh-hannover.de (T.G.); 3Department of Cardiac-, Thoracic-, Transplantation and Vascular Surgery, Hannover Medical School, 30625 Hannover, Germany; 4Institute of Clinical Biochemistry, Hannover Medical School, 30625 Hannover, Germany; 5Proteomics, Institute of Theoretical Medicine, Faculty of Medicine, University of Augsburg, 86159 Augsburg, Germany; 6Biomedical Research in End Stage and Obstructive Lung Disease Hannover (BREATH), German Center for Lung Research (DZL), 30625 Hannover, Germany

**Keywords:** xenotransplantation, endothelial cells, glycan oxidation, xenoantigens, pig

## Abstract

Background: Patterns of humoral immune responses represent a major hurdle in terms of pig-to-human xenotransplantation approaches. The best-known xenogeneic glycan antigens present in pigs are the αGal (Galili antigen) and the non-human sialic acid Neu5Gc. As there are further differences between porcine and human cellular surface glycosylation, a much broader range of glycan epitopes with xeno-reactive relevance can be anticipated. Therefore, we set out to chemically modify porcine cellular surface glycans in a global approach by applying sodium periodate (NaIO_4_) oxidation. Methods: Porcine endothelial cells were exposed to oxidation with 1 to 5 mM NaIO_4_ for different time periods at 37 °C or 4 °C and under static or dynamic conditions. The impact on cellular survival was determined by applying live/dead assays. Oxidation of αGal-epitopes was assessed by fluorescence microscopy-based quantification of isolectin-B4 (IL-B4) staining. Overall immunogenicity of porcine cells was determined by human serum antibody binding. Results: Treatment of porcine endothelial cells and tissues with NaIO_4_ led to reduced binding of the αGal-specific IL-B4 and/or human serum antibodies. NaIO_4_ was revealed to be cytotoxic when performed at elevated temperatures and for a prolonged time. However, by applying 2 mM NaIO_4_ for 60 min at 4 °C, a high extent of cellular viability and a relevant reduction in detectable αGal epitope were observed. No differences were detected irrespectively on whether the cells were oxidized under static or flow conditions. Conclusions: Glycan epitopes on living cells can be oxidized with NaIO_4_ while maintaining their viability. Accordingly, this strategy holds promise to prevent immune reactions mediated by preformed anti-glycan antibodies.

## 1. Introduction

End-stage organ failure threatens the lives of millions of affected people worldwide and tremendously diminishes their quality of life. It poses a profound challenge, with organ transplantation often being the only available long-term treatment option. However, there remains a substantial gap between the demand for respective donor organs and their availability [1,2,3].

To address this pressing issue, one promising avenue is xenotransplantation.

Nevertheless, the immunological mismatch between donor and recipient species is yet one of the most considerable hurdles to overcome with regard to possible future xenotransplantation approaches. Scenarios of different mechanisms of graft rejection, including hyperacute graft rejection (HAR), which leads to immediate graft failure upon transplantation, are relevant consequences of species-related immunological differences [4]. HAR represents a fast and very strong immune reaction mediated by the binding of preformed antibodies directed to particular xenoantigens present on the cellular surfaces of transplanted organs and tissues. With respect to pig-to-human xenotransplantation settings, surface glycans especially play a pivotal role in early graft rejection. Several glycan structures with a relevant meaning in terms of pig-to-human xenotransplantation are known today, i.e., αGal, Neu5Gc, and Sda. As in the nature of things, binding of preformed and induced antibodies primarily occurs on endothelial cells (ECs) lining the vasculature of the transplant, which leads to subsequent complement activation and cell-mediated cytotoxicity, EC activation, and different vascular effects like thrombosis and vasoconstriction, finally resulting in early organ failure [5,6].

Today’s most common approach in order to prevent HAR in pig-to-primate xenotransplantation is based on the utilization of organs and tissues derived from specific genetically engineered donor pigs. Enzymes involved in the synthesis of the three known porcine xenogeneic surface glycans (αGal, Neu5Gc, and Sda) are genetically knocked out, leading to an overall reduced binding of human IgG- and IgM-antibodies to respective porcine tissues and cells [7,8].

However, recent investigations have shown that even tissues derived from respective triple knock-out pigs still possess particular immunogenicity in contact with the human immune system [9], suggesting the existence of additional xenoantigenic epitopes on porcine endothelial cells (PECs), which are targeted by preformed or at least induced human antibodies [9,10,11,12]. As these structures are not yet identified, the respective genetic engineering of potential donor pigs is not feasible.

In 2003, De Bank et al. described a method to label and extract cell-surface sialic acid-containing glycans from living animal cells [13]. This method used mild concentrations of sodium periodate (NaIO_4_) to selectively introduce an aldehyde at the glycerol side chain of sialic acids [14]. At higher concentrations, NaIO_4_ can also oxidize vicinal diols within the cyclic sugar backbone, enabling the oxidation of a broader spectrum of glycans [15]. Therefore, the NaIO_4_ treatment is likely to also chemically modify immunogenic glycans. Thus, NaIO_4_ oxidation represents a promising strategy in xenotransplantation for the modification of cells and tissues with the aim to destroy immunogenic glycan epitopes against which preformed antibodies might exist within the recipient. In this study, we used PECs and porcine tissues to investigate the suitability of NaIO_4_ treatment to destroy cellular glycan antigens using αGal as a well-known epitope and investigated the effect on the viability of living PECs.

## 2. Materials and Methods

### 2.1. Isolation of Tissue Samples from Porcine Thoracic Aorta

Pigs were bred at the Ferkelerzeugergemeinschaft Coppenbrügge and delivered to the central animal facility of Hannover Medical School. Explantation of the tissue samples was performed from euthanized animals in an animal operating room under sterile conditions. All pigs included in this study were German landrace crossed with Duroc females and weighed between 31 and 40 kg prior to explantation. Retrieval of aortic sections was conducted secondary to lung procurement, which was the primary reason for the animal experiment and had been approved by an external ethics committee (Lower Saxony State Office for Consumer Protection and Food Safety, LAVES, Oldenburg, Germany). Aortic sections with a length of 10 cm were dissected from the descending part of the thoracic aorta. After dissection, tissues were stored for a maximum of 4 h in Ringer’s solution at 4 °C.

### 2.2. Isolation of PECs from Tissue Samples

Connective tissue was removed from the aorta. Afterward, aortic sections were rinsed in PBS in order to remove tissue debris. Subsequently, aortic sections were cut open lengthwise and placed on top of a 12-well lid with the luminal side up. In order to allow cellular isolation from a specific area, a 3D-printed form defining an area of 20 mm × 40 mm was clamped onto the vessel with spring clamps. Detachment of PECs was conducted by application of 7 mL of collagenase Type 2 (650 U/mL) in PBS (Worthington Biomedical, Lakewood, NJ, USA) onto the vessel area for 8 min at room temperature. Afterward, collagenase solution containing the detached cells was collected and mixed with Medium 199 (Lonza, Walkersville, MD, USA) containing 10% Fetal Calf Serum (FCS) (PAN Biotech, Aidenbach, Germany) in a 1:1 ratio. Cell scrapers were used in order to support cellular detachment after collagenase treatment. Scrapers were rinsed with Medium 199 containing 10% FCS. In order to maximize the number of isolated cells, luminal vessel sides were washed again with PBS, which was collected thereafter as a third fraction. All three preparations were then centrifuged for 5 min at 400× *g* and supernatants were discarded. Cell pellets were resuspended in 5 mL of EGM-2 (Lonza) with an additional 5% FCS and the cells were subsequently seeded onto separate T-25 cell culture flasks and cultured at 37 °C, 5% CO_2_.

### 2.3. Cell Monolayer Microfluidic System

The BioFlux one microfluidic system (Fluxion Biosciences, Alameda, CA, USA) was used to assess the efficiency of chemical oxidation under dynamic flow properties in order to closely emulate physiological conditions. Initially, 48-well microflow plates were coated with 100 μg/mL of fibronectin (Invitrogen, Carlsbad, CA, USA) for 1 h at 37 °C. Channels were washed thereafter with EGM-2 at a shear flow of 15 dyne/cm^2^ and seeded with 10 µL of PECs (135,000 cells) in EGM-2 through the inlet well. After 1 h of incubation at 37 °C, plates were filled with fresh EGM-2 and further incubated for 24 h before use.

### 2.4. Solubility of Sodium Periodate

Solutions of 10 mM NaIO_4_ (Honeywell Fluka, Seelze, Germany) were prepared in PBS, EBM-2 (Lonza), or isotonic saline (0.9% NaCl) and further diluted to 1 mM, 2 mM, 3 mM, 4 mM, and 5 mM in the respective solvent. Prepared solutions were stored at 4 °C for 24 h. After the cooling period, all samples were checked for the appearance of precipitations.

### 2.5. Oxidation of PECs with NaIO_4_

For oxidation under static conditions, PECs were seeded into 24-well plates and cultivated as described above until reaching confluency. Oxidation of PECs was performed either at 4 °C or 37 °C. Plates were placed onto a tumbling shaker at a speed of 25 rpm during the procedure. At first, the cells were washed once with PBS; afterward, 500 µL of NaIO_4_ solution (final concentration of 1–5 mM in 0.9% NaCl) was applied to the cells. After respective oxidation time, the NaIO_4_ solutions were discarded and the cells were washed once with cold PBS. Depending on further investigations, the cells were cultured in order to perform live/dead analysis or fixed for staining. PECs treated only with PBS were used as untreated controls, whereas PECs incubated with 70% methanol for 30 min at 37 °C were used as negative controls for the live/dead assays.

For investigation of dynamic flow conditions, the cells were seeded into channels of 48-well plates (Fluxion Biosciences, Alameda, CA, USA) in a microfluidic system capable of generating shear stresses ranging from 0 to 200 dyne/cm^2^. Well plates to be processed were placed on a cooling plate throughout the procedure in order to maintain a constant temperature of 4 °C or placed into the BioFlux heating plate at 37 °C. After rinsing the cells with HBSS using a shear flow of 10 dyne/cm^2^, the cells were treated with NaIO_4_ solutions at concentrations of 1–3 mM for 60 min under flow conditions (5 dyne/cm^2^ for 10 min, 40 min gravity flow, and 5 dyne/cm^2^ for 10 min). The controls were treated with PBS only. Post-oxidation, the cells underwent washing with PBS twice and were either fixed for staining using 4% paraformaldehyde (PFA) in PBS or replenished with EGM-2 and cultivated for 24 h. Human endothelial colony-forming cells (ECFCs), kindly provided by Michael Pflaum [16], were used as negative controls for IL-B4 and serum staining and treated similarly to the PBS controls.

### 2.6. Live/Dead Analysis

After periodate oxidation, the cells were incubated at 37 °C and 5% CO_2_ with EGM-2 cell culture medium for 24 h. The culture medium was removed and the cells were washed with PBS. LIVE/DEAD™ Viability/Cytotoxicity assay (Invitrogen) was performed according to the manufacturer’s instructions. In the microfluidic system, the cells were washed once with PBS employing a shear flow of 15 dyne/cm^2^. Subsequently, 100 μL of Calcein AM and ethidium homodimer were added to the inlet well followed by flushing the channels with a shear flow of 5 dyne/cm^2^ for 5 min.

### 2.7. Isolectin-B4 (IL-B4) Staining

Immediately after oxidation, the cells were fixed with 4% PFA for 20 min at room temperature. After washing with PBS, the cells were blocked with 1% bovine serum albumin (BSA) (Sigma-Aldrich, St Louis, MO, USA) in PBS for 1 h at room temperature followed by incubation with Isolectin-B4-Cy5 (DL-1208, Vector Laboratories Inc., Burlingame, CA, USA) diluted 1:200 in PBS at 4 °C overnight. Subsequently, the cells were washed with PBS at room temperature and nuclei were stained using DAPI.

In the microfluidic system, the cells were fixed using 4% PFA for 5 min applying a shear flow of 15 dyne/cm^2^. The fixed cells were blocked for 1 h with 1% BSA solution and subsequently incubated with 1:200 diluted Isolectin-B4-Cy5 overnight at 4 °C. Afterward, channels were washed with PBS under a shear flow of 5 dyne/cm^2^. Finally, channels were flushed with 2 drops of Mount FluorCare DAPI (Carl Roth, Karlsruhe, Germany) solution applying a shear flow of 10 dyne/cm^2^ for 5 min in order to facilitate nuclear staining.

### 2.8. Binding of Human Serum

Immediately after oxidation under dynamic conditions, the cells were exposed to human serum (blood group O) for 1 h applying a shear flow of 20 dyne/cm^2^. Afterward, the cells were fixed with 4% PFA employing a shear flow of 15 dyne/cm^2^ for 5 min, and then subsequently blocked with 500 μL 1% BSA solution, employing a shear flow of 15 dyne/cm^2^ for 1 h. After washing with 1× TBS employing a shear flow of 15 dyne/cm^2^ for 10 min, human antibodies were detected by applying a FITC-conjugated goat anti-human IgA, IgG, IgM (Heavy and Light Chain) antibody (ABIN100791, antibodies-online GmbH, Aachen, Germany) for 1 h using gravity flow. Supernatant antibodies were removed by rinsing with 1× TBS under a shear flow of 15 dyne/cm^2^ for 2 min. Finally, channels were flushed with two drops of DAPI solution using a shear flow of 10 dyne/cm^2^ for 5 min to facilitate nuclear staining.

### 2.9. VE-Cadherin Staining

The PFA-fixed cells were washed with 500 µL PBS and blocked with 2% donkey serum in PBS for 60 min at room temperature. After discarding the blocking solution, 200 µL of 1:100 diluted rabbit anti-human CD144 antibody (AHP628Z, Bio-Rad AbD Serotec, Neuried, Germany) and 1:100 diluted rabbit IgG-Isotype control in PBS (Abcam, Cambridge, MA, USA) were applied at 4 °C overnight. Subsequently, the cells were washed with 500 µL PBS and incubated with 200 µL of a 1:300 diluted Cy3-donkey-anti-rabbit IgG (Bio-Rad AbD Serotec, Neuried, Germany) for one hour at room temperature. Afterward, the cells were washed with 500 µL of PBS and finally covered with 500 µL of PBS.

### 2.10. Analysis of the Penetration Depth

Thoracic aortic sections were prepared as described above. Aortic sections were then washed with 6 mL of 4 °C cold PBS prior to the application of 6 mL of 2 mM NaIO_4_ solution for 40 min at 4 °C. The NaIO_4_ solution was discarded thereafter and the luminal sides of the aortic sections were subsequently washed again with 6 mL of PBS at 4 °C. Tissue pieces of treated aortic sections were embedded in Tissue-Tek O.C.T. ^TM^ (Sakura Finetek, Alphen aan de Rijn, The Netherlands) and snap-frozen over liquid nitrogen. Cryosections of 5 µm thickness were prepared using a standard cryotome. Cryosections were fixed with 100 µL of 4% PFA for 15 min at room temperature. Subsequent IL-B4 staining was performed using the same protocol as described above.

### 2.11. Quantification of Live/Dead Assays

The quantification of living and dead cells was performed on PECs seeded in well plates by using images taken with a microscope (Axio Observer A1, Zeiss, Oberkochen, Germany) using the exact same setting and magnification for each picture. ImageJ2 (Version 2.3.0) image analysis software with the Fiji plugin (Version 2.3.1)was used to quantify images derived from conducted live/dead assays. On that account, RGB channels were split and the red and green channels were analyzed separately. After adjustment of brightness, respective images were processed to binary pictures choosing an appropriate threshold. Binary dots, each representing a cell, were then watershed in order to separate agglomerates of cells. Afterward, automatized dot counting was performed by ImageJ2 software with the Fiji plugin. Respective survival rates (% survival) were finally calculated as follows:$$survival\;rate\;[\%]=\frac{n\left(living\;PEC\right)\;*\;100}{n\left(living\;PEC\right)+n\;(dead\;PEC)}$$

### 2.12. Statistics

If not stated otherwise, experiments were performed in biological triplicates using materials from 3 different animals. Statistical analysis was performed by two-way ANOVA followed by Bonferroni post-tests using GraphPad PRISM software (Version 10.1.0). A measured difference was considered significant with a *p*-value lower than 0.05 (*), 0.01 (**), 0.001 (***), or 0.0001 (****).

## 3. Results

### 3.1. Isolated Cells Express EC Markers

PECs were isolated from the thoracic aorta of three different pigs using a standard procedure for the isolation of ECs. The cells derived from all three isolations exhibited a typical endothelial cell cobblestone morphology in culture (Figure 1A). Staining against VE-cadherin revealed a cell surface localization of this EC marker protein on the cultured cells (Figure 1B). The obtained cells were also positive upon staining with the lectin IL-B4 (Figure 1C), indicating the expression of αGal-epitopes. Together, these analyses confirmed the EC phenotype of the isolated cells.

### 3.2. NaIO_4_ Can Be Dissolved in Isotonic Saline at Sufficient Concentrations

Periodate oxidation of living cells at 37 °C has been described by NaIO_4_ dissolved in PBS [13]. In our experimental setup, different temperature conditions were tested during periodate oxidation. We noted that upon cooling a NaIO_4_ solution in PBS to 4 °C, precipitation of a white substance was observed even at 2 mM NaIO_4_. The amount of precipitate augmented with increasing NaIO_4_ concentration, implying that NaIO_4_ is not fully soluble in 4 °C cold PBS at concentrations above 2 mM (Figure 2A). In contrast, no precipitation was observed at 4 °C up to 5 mM NaIO_4_, when EBM-2 or isotonic saline was used as a solvent (Figure 2B,C). As the EBM-2 medium contains glucose, which is likely to react with NaIO_4_, we decided to apply saline as a solvent for NaIO_4_ for all further experiments.

### 3.3. Cell Survival Can Be Optimized upon Oxidation at Low Temperature

PECs were treated with 2 mM NaIO_4_ under static conditions for 40 min at either 4 °C or 37 °C. Cell viability was assessed 24 h after treatment by live/dead assay. Treatment at 4 °C resulted in 93.0 ± 5.4% viable cells (Figure 3A,C), while only 0.09 ± 0.03% of PECs were viable after treatment conducted at 37 °C (Figure 3A,F) compared to a 100% survival of untreated cells at both temperatures (Figure 4A,B,E). Under dynamic conditions, no viable PECs could be found after oxidation with 2 mM NaIO_4_ at 37 °C (Figure 3G), whereas most cells were viable when oxidized with 2 mM NaIO_4_ at 4 °C (Figure 3D).

### 3.4. Increasing Oxidation Time and NaIO_4_ Concentration Negatively Affect Cell Survival

PECs were treated under static conditions at 4 °C with different concentrations of NaIO_4_ (2–5 mM) and increasing durations (10–60 min). Survival rates of PECs were determined using a live/dead assay. While cellular viability is maintained at 100% upon treatment with 2 mM NaIO_4_ even after prolonged treatment up to 60 min, higher concentrations of 3 mM or 4 mM NaIO_4_ considerably decrease the survival of PECs when treated for more than 30 min (Figure 4A).

At 5 mM NaIO_4_, the viability of PECs was already reduced after 20 min of treatment; after 60 min, survival rates of only 4.3 ± 7.2% were observed (Figure 4A).

### 3.5. Oxidation of PECs with NaIO_4_ Reduces Detection of αGal by IL-B4

Based on our previous analyses, we defined concentrations of periodate and times for treatment at which PECs survived at high rates which were 2 mM NaIO_4_ for 60 min, 3 mM NaIO_4_ for 30 min, 4 mM NaIO_4_ for 20 min, and 5 mM NaIO_4_ for 10 min at 4 °C (Figure 4B,E,H,K). These conditions were chosen for the analysis of IL-B4 binding to treated PECs (Figure 4C,F,I,L). IL-B4 binding to the cell surface as deduced from staining of cell borders was considerably reduced in cells oxidized with 2 mM NaIO_4_ for 60 min compared to untreated controls (Figure 4C,D). The other depicted settings revealed only mild or no additional reduction in the extent of IL-B4-positive staining (Figure 4F,G,I,J,L,M).

### 3.6. Binding of IL-B4 and Human Serum Antibodies to Cells Is Reduced after Oxidation under Dynamic Conditions

In addition to treatment under static conditions, PECs were treated with NaIO_4_ under dynamic conditions employing the BioFlux system. Similar to oxidation under static conditions, the cells showed a reduction in IL-B4 staining after oxidation with 2 mM NaIO_4_ for 60 min at 4 °C under dynamic conditions (Figure 5A,B,E,F). Oxidation with 3 mM NaIO_4_ for 60 min further reduced the IL-B4 staining (Figure 5C,G). Similar observations were made with human serum, which showed reduced binding of human antibodies to PECs oxidized for 60 min with 2 mM (Figure 5I,J,M,N) and a further reduction with 3 mM NaIO_4_ (Figure 5K,O). Human ECFCs served as negative controls and exhibited even less staining (Figure 5D,H,L,P).

### 3.7. αGal Epitopes on Cell Surfaces Undergo a Quick Turnover

In order to investigate the glycan turnover on EC membranes, PECs were cultivated after the oxidation process for 24 h and subsequently stained with IL-B4. Staining results were compared to the corresponding staining of PECs derived from the same isolations directly stained after oxidation as carried out in previous experiments (Figure 6). PECs stained directly after oxidation exhibited no staining (Figure 6A), whereas PECs cultivated for 24 h after oxidation exhibited diffuse staining with pronounced accentuation of cellular membranes (Figure 6B). The observed intensity of IL-B4 staining did not differ from respective untreated controls (Figure 6C,D), indicating a mostly complete turnover of previously oxidized glycans.

### 3.8. NaIO_4_ Can Oxidize Glycans at Deeper Layers upon Diffusion into Native Aortic Tissue

Based on the envisioned application of periodate oxidation for the treatment of tissues intended for xenotransplantation, native aortic tissue was used to determine if NaIO_4_ treatment also reduces IL-B4 staining in tissues_._ Luminal surfaces of native porcine aortic sections were treated with 2 mM NaIO_4_ for 40 min at 4 °C. Upon staining of tissue sections with IL-B4, it could be shown that periodate was capable of diffusing into the media layers of investigated aortic vessel walls (Figure 7A). Staining of endothelial layers appears to be slightly brighter in comparison to underlying vessel layers, but the detected intensity of staining was clearly reduced compared to the unoxidized controls (Figure 7B).

## 4. Discussion

HAR and acute vascular rejection (AVR) of porcine organs and vascularized tissues are mediated by antibodies binding to xenoantigens on ECs coating the graft-related vessels. Since most of the xenoantigens currently known and characterized are carbohydrates, their removal or alteration towards non-immunogenic structures are crucial steps in order to reduce the immune response of the human recipient immune system upon exposure to a vascularized porcine transplant [9]. In this study, we showed that recognition of carbohydrate xenoantigens can be mitigated using simple periodate oxidation, without influencing the vitality of the carrying ECs. By application of this method on the endothelium of vascularized transplants or blood-exposed surfaces in general, hyperacute or acute rejection characteristics of a human immune system towards porcine tissues or organs might be prevented or at least reduced to insignificant extents.

Other strategies to reduce the immunogenicity of porcine organs involve the genetic knock-out of three known xenoantigens (αGal, Neu5Gc, and Sda) and the expression of complement regulatory proteins such as CD46, CD55, or CD59 (reviewed in [17]). Comparable to the NaIO_4_ oxidation, the binding of preformed antibodies to porcine cells can be reduced by the knock-outs [5]. Because knock-outs are not sufficient to completely abolish complement activation and deposition, additional human complement regulatory proteins are expressed [18,19]. The reason for the remaining low levels of complement activation could be the presence of not yet identified xenoantigens, which might be targeted by the NaIO_4_ treatment.

### 4.1. Porcine Endothelial Cells as Target Cells

In this study, we combined a classical standard protocol for EC isolation with a custom-made device to isolate PECs. Thus, the isolated cells exhibited the typical cobblestone endothelial morphology and stained positive for VE-cadherin and IL-B4, indicating an almost pure population of ECs [13,20]. ECs have a particular importance with respect to every transplantation scenario including the transplantation of a vascularized graft since they are the first cells of the donor that make contact with the recipient’s blood flow, which has eminent relevance in terms of xenotransplantation purposes.

A distinct limitation of this study is the use of only one type of EC, in particular aortic-derived ECs. ECs comprise heterogeneous cell populations with varying properties among each other, depending on their location in the circulatory system [20]. This variability might also affect their susceptibility to oxidative stress and, similarly, the properties of their cell surface glycans may differ. Therefore, deduced predictions with respect to other compartments of the porcine circulatory system or whole organs can only be made very carefully and most notably require further investigations.

### 4.2. Oxidation Conditions

Previous studies describe the solubilization of NaIO_4_ in PBS prior to application as an oxidant for cell surface glycans [13]. However, in our study, we observed precipitates when NaIO_4_ was solubilized in PBS already at concentrations of 2 mM at 4 °C. Most likely, observed precipitations consisted of KIO_4_, which is formed upon reaction with potassium ions present in PBS and is poorly soluble in aqueous solutions [14]. This prompted us to investigate other solvents for NaIO_4_. Cell culture media, like EBM-2 medium, which would be an ideal diluent for our purposes, exhibited no precipitation with NaIO_4_ (Figure 2). However, we decided not to use EBM because NaIO_4_ might react with ingredients of the medium, e.g., glucose and proteins, thereby altering the effective NaIO_4_ concentration. Furthermore, toxic metabolites might be formed as well, which naturally could also relevantly interfere with the achievable results of the experiments conducted in this work. In order to generate reliable concentrations of NaIO_4_ and avoid toxic metabolites, we used saline solution for dissolving NaIO_4_, which did not lead to any precipitations, even at a high concentration of 5 mM NaIO_4_ and a low temperature of 4 °C (Figure 2). For future experiments, the solvent of choice for NaIO_4_ could be optimized by including components that facilitate cell survival but do not interfere with the reaction of NaIO_4_.

Additionally, we observed a clear difference in cell survival depending on the temperature at which respective oxidations were conducted. Periodate oxidation at 4 °C significantly improved cellular survival rates in comparison to oxidations conducted at 37 °C. In previous studies, a high level of cytotoxicity of Madin–Darby bovine kidney (MDBK), A549, and Vero cells treated with up to 100 mM NaIO_4_ at 37 °C for 1 h could be observed, but not when treated with 10 mM [21]. In our study, 2 mM NaIO_4_ was already cytotoxic which might highlight that primary cells are much more sensitive to NaIO_4_ oxidation compared to stable cell lines. In this study, we did not perform experiments to elucidate the underlying reasons and mechanisms, but we speculate that due to lower temperatures, the uptake of toxic products such as the NaIO_4_ itself or already formed aldehydes by the ECs might be reduced. Previous studies suggest that endocytosis can be reduced to a minimum at 4 °C [22]. A major drawback of periodate oxidation conducted at 4 °C is the considerably slower reaction kinetics when compared to respective oxidations performed at 37 °C. In order to achieve similar effectiveness at lower temperatures, the incubation time has to be prolonged, which again might cause other relevant side effects. With regard to potential application strategies in the context of intended xenotransplantation scenarios, oxidation at 4 °C can be easily combined with cold organ perfusion, which was recently introduced to be highly effective at least when applied in xenogeneic heart transplantations [23].

### 4.3. Xenogenic Epitopes

In our current work, we showed a distinct reduction in IL-B4 binding to porcine tissues and PECs after oxidation with NaIO_4_, indicating that the αGal epitope, which is recognized by IL-B4, was altered in a way that subsequently hampered the binding capacity of this lectin. In addition, the binding of human sera to PECs oxidized by NaIO_4_ was reduced as well. Therefore, it can be assumed that periodate-mediated oxidation of αGal prevents its recognition by preformed human antibodies as well. Furthermore, based on the broad glycan spectrum that is affected by the oxidation process, it can also be anticipated that other glycan epitopes, to which an immunologically relevant antibody binding is expected to occur upon transplantation, might be rendered to be no longer detectable after this procedure.

However, as described by previous investigations, approximately 40–55% of ^3^H-labeled glycoproteins from cells subjected to periodate oxidation and aniline-catalyzed oxime ligation were still detectable by subsequent biotin labeling [14]. This suggests that not every glycan was modified during the described processing. Conclusively, we observed likewise that oxidized PECs still exhibited IL-B4 detection in particular areas surrounding cellular nuclei. Most likely, this can be explained by positive IL B4 detection of unoxidized glycans present in the Golgi apparatus and other intracellular compartments. Interestingly, these glycans seemed to be protected from the oxidation process conducted in this study and thus could replace oxidized glycans on the cellular surface in the time period after the introduced procedure. Indeed, we observed a recovery of structures positively detectable by IL-B4 staining within 24 h after initial oxidation, indicating the replacement of oxidized glycans with de novo and thus unoxidized glycans in that time frame. As a crucial consequence, the therapeutic window of the beneficial effects of glycan oxidation might be very short and thus is only capable of reducing the severity of early-phase hyperacute and acute immune reaction patterns during the first hours after transplantation. On the other hand, the strategy of immune modulation introduced in this study might be very useful for permanently changing potentially xeno-reactive glycans retained in metabolically inactive porcine-derived biological implants, for example, glutaraldehyde-fixed bioprosthetic heart valves.

### 4.4. Oxidation under Shear Flow Conditions

Experiments were performed in this work under shear flow with the intention to immediately remove toxic byproducts of the oxidation from processed cellular surfaces and by this means to improve cell viability. However, respective experiments included in this study revealed no relevant impact of flow conditions on cellular survival, suggesting that toxic byproducts might not be a notable cause of decreased cellular viability. One alternative explanation might be a direct cellular uptake of NaIO_4_ causing deleterious intracellular damages and/or changes in the cellular membranes that are induced by NaIO_4_ oxidation. During the same experiments, live/dead assays were performed immediately after oxidation, revealing cells stained positive for Calcein AM and for nuclear staining as well, which is indicative of damaged cellular membranes.

### 4.5. Tissue Penetration Depth

ECs mediate the first contact between the recipient human immune system and a transplanted vascularized xenograft, but shortly after transplantation human cells migrate into the tissue of the donor organ as well [4]. Therefore, we additionally investigated if NaIO_4_-mediated oxidation could be applied in porcine donor tissues beyond vasculature as well. Thus, we observed that NaIO_4_ is capable of penetrating into the examined tissues, even without perfusion through capillaries. This might be relevant in terms of oxidizing whole xenogeneic donor organs in order to further reduce pre-existing immunologic burdens.

### 4.6. Side Effects

In the current study, we focused our analyses on xenoantigens present in endothelial cells. However, our findings upon oxidation of tissues suggest that other cell types are oxidized as well. Multiple different effects of NaIO_4_ treatment such as the inhibition of migration of macrophages or a dramatic increase in platelet aggregation have been described in other studies [24,25]. The underlying cause could be the changes in the intermolecular interaction of glycans or glycosylated molecules, for example, in cell metabolism, membrane integrity, or receptor–ligand interactions that are all influenced by the oxidation of vicinal diols [26]. Therefore, in addition to vitality, changes in cellular behavior should be considered in future experiments.

### 4.7. Further Investigations

After demonstrating the proof of principle on PEC monolayers in this study, it remains a crucial necessity to further investigate both immunogenicity and biocompatibility of oxidized PECs as well. A rational next step would be to study the influence of NaIO_4_ on other cell types and cell-to-cell interaction, for example, with macrophages and other immune cells. In order to improve cellular survival and to expand the therapeutic margin of the techniques introduced in this study, the composition of the oxidation solution to be applied should be further refined, including cell-protective substances such as ascorbic acid. Another future step in order to further evaluate the potential utility of glycan oxidation approaches is whole organ perfusion of potential porcine donor organs using ex vivo perfusion systems and strategies.

## 5. Conclusions

We developed a NaIO_4_ oxidation protocol that allows chemical modification of glycans impairing their recognition of living PECs by lectins or antibodies.

## Figures and Tables

**Figure 1 biomedicines-12-01470-f001:**
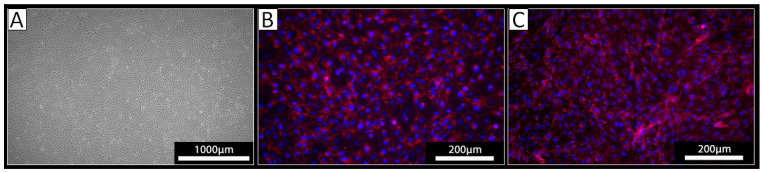
Analysis of isolated PECs. (**A**) Light microscopy revealed a cobblestone-like growth pattern of the isolated cells. (**B**,**C**) Immunocytochemistry of the isolated cells using (**B**) an antibody against VE-Cadherin (red) or (**C**) the lectin IL-B4 against αGal (red); nuclei were stained with DAPI (blue).

**Figure 2 biomedicines-12-01470-f002:**
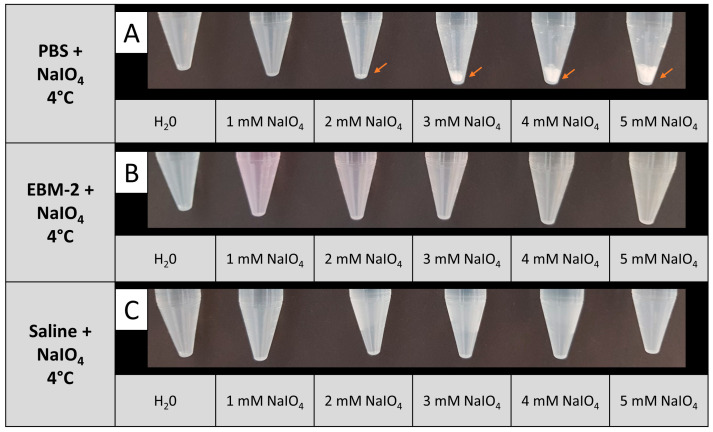
Solubility of NaIO_4_ at 4 °C in different solvents. NaIO_4_ was dissolved in PBS (**A**), EBM (**B**), or saline (**C**) at the indicated concentrations. In PBS, the formation of a white precipitate was observed starting from a concentration of 2 mM NaIO_4_ (orange arrows); n = 2.

**Figure 3 biomedicines-12-01470-f003:**
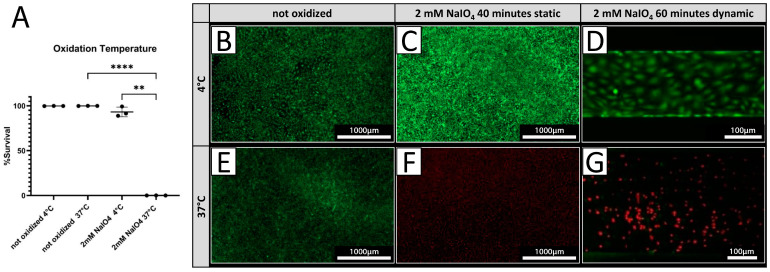
Cell viability is dependent on oxidation temperature. (**A**) Survival rates of PECs oxidized with 2 mM NaIO_4_ for 40 min at 4 °C or 37 °C with respective controls incubated with PBS (n = 3). (**B**–**G**) Exemplary pictures of a live/dead assay (green = live; red = dead) of PECs treated at 4 °C (**B**–**D**), respectively, and at 37 °C (**E**–**G**). (**C**,**F**) NaIO_4_ oxidation under static conditions was compared to oxidation under dynamic conditions. (**D**,**G**) Experiments were performed with cells from three different donor animals. Scale bar 1000 µm in (**B**,**C**,**E**,**F**) or 100 µm in (**D**,**G**). *p*-values of ≤ 0.01 (**), respectively ≤ 0.0001 (****).

**Figure 4 biomedicines-12-01470-f004:**
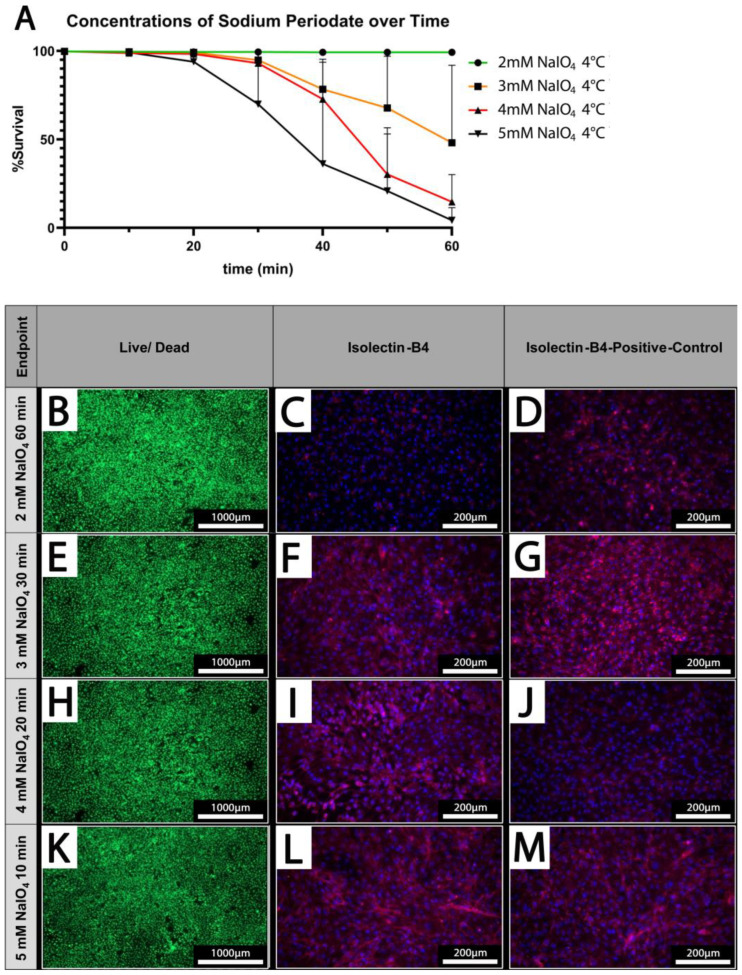
Cell survival is dependent on applied concentrations and treatment duration. (**A**) The cell survival rate at different concentrations of NaIO_4_ over time; (n = 3). (**B**,**E**,**H**,**K**) Live/dead staining of PECs oxidized using 2–5 mM NaIO_4_ for 60, 30, 20, or 10 min. (**C**,**F**,**I**,**L**) IL-B4 staining of PECs treated with the indicated different concentrations of NaIO_4_ and duration. (**D**,**G**,**J**,**M**) Respective controls; PECs treated with PBS for the indicated periods. Scale bars show (**B**,**E**,**H**,**K**) 1000 µm or (**C**,**D**,**F**,**G**,**I**,**J**,**L**,**M**) 200 µm.

**Figure 5 biomedicines-12-01470-f005:**
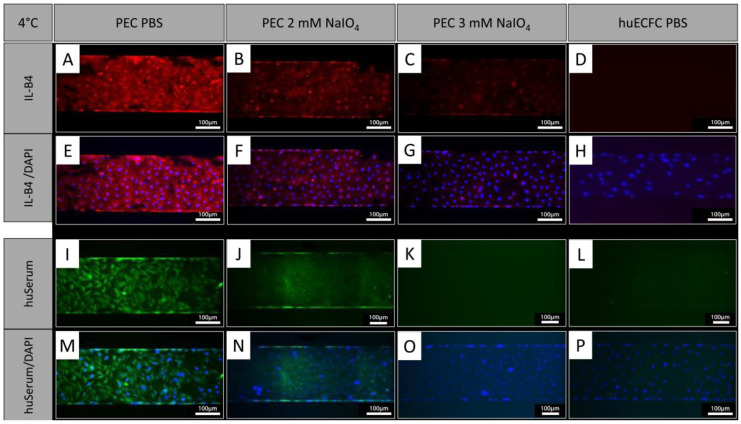
NaIO_4_ oxidation under dynamic conditions. (**A**,**E**,**I**,**M**) PECs treated with PBS as controls and stained with IL-B4 or human serum. (**B**,**F**) Oxidation for 60 min with 2 mM NaIO_4_ caused a similar reduction in IL-B4 staining as static conditions (blue = DAPI; red = IL-B4). (**C**,**G**) Even stronger reduction was observed using 3 mM NaIO_4_ for 60 min. (**I**–**P**) Staining of human IgG, IgA, and IgM showed reduced binding of human serum antibodies to PECs oxidized for (**J**,**N**) 60 min with 2 mM and a further reduction with (**K**,**O**) 3 mM NaIO_4_ (blue = DAPI, green = anti-human IgA, IgG, and IgM (Heavy and Light Chain)). (**D**,**H**,**L**,**P**) Human ECFCs served as negative controls. Experiments were performed in technical triplicates with cells from one donor animal. Scale bar: 100 µm.

**Figure 6 biomedicines-12-01470-f006:**
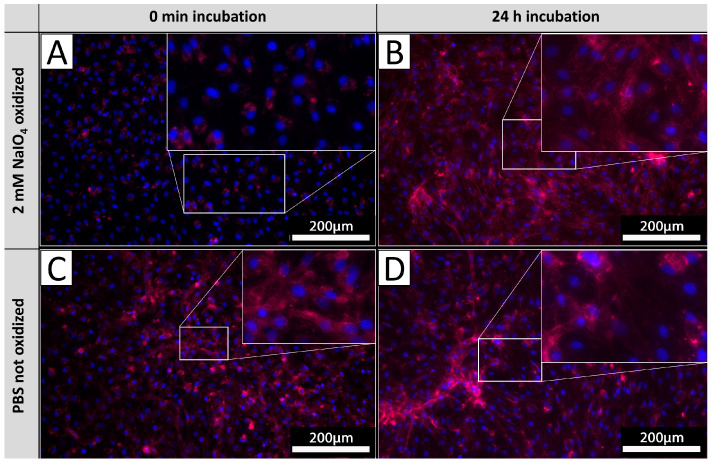
αGal turnover. PECs stained with IL-B4 (red) directly after oxidation (**A**) and corresponding untreated controls (**B**). PECs cultivated for 24 h after oxidation and stained with IL-B4 (**C**) and respective unprocessed control (**D**). The extent of areas positively detected by IL-B4 staining increases on levels of unoxidized tissues after 24 h of further incubation. Experiments were performed as biological duplicates with cells from two independent donor animals. Scale bar: 200 µm.

**Figure 7 biomedicines-12-01470-f007:**
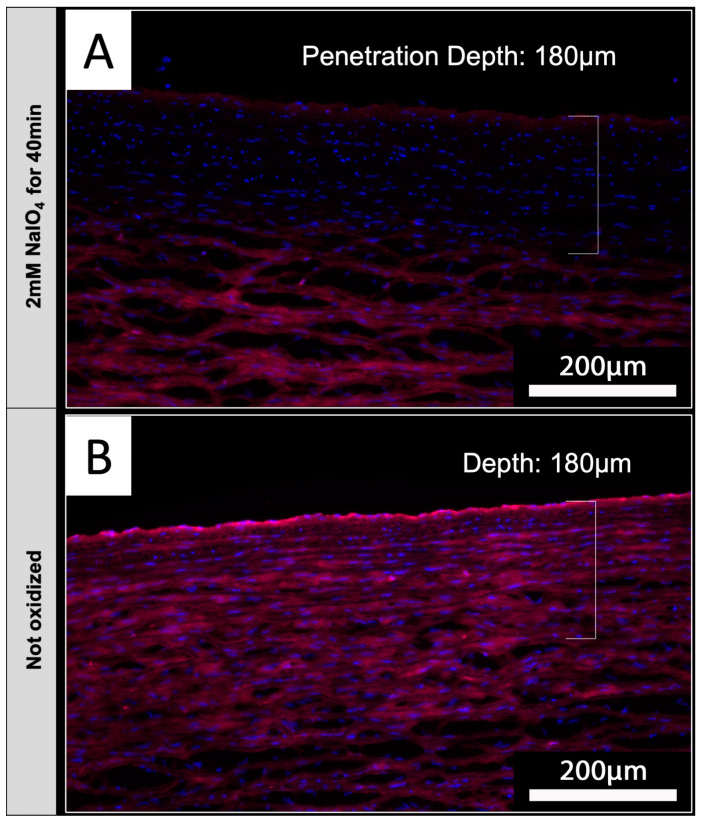
NaIO_4_ penetrates into native aortic tissue. Cross-sections of treated (**A**) and untreated (**B**) aorta were stained with IL-B4. (**A**) Oxidation with 2 mM NaIO_4_ for 40 min at 4 °C led to a distinct reduction in IL-B4 positivity (red) within the intima layer and parts of the media layer. Cellular nuclei were counterstained with DAPI (blue). (**B**) Complete media layer and even more the intima layer of untreated aorta exhibit IL-B4 staining. This experiment was performed using tissue from only one animal. Scale bar: 200 µm.

## Data Availability

The original contributions presented in this study are included in the article. Further inquiries can be directed to the corresponding author.

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
