# Peer review of "Reduction in Xenogeneic Epitopes on Porcine Endothelial Cells by Periodate Oxidation"

_biomedicines, 2024, doi:10.3390/biomedicines12071470_

Round 1
Reviewer 1 Report
Comments and Suggestions for Authors
Dear authors, thank you very much for allowing me to read your work. You describe a method of quenching endothelial cell epitopes (porcine endothelial cells originating from aortic tissue) with the use of sodium periodate. Since Xenotransplantation has real-life applications your work is relevant. I have the following comments:
Sodium periodate has been known to modulate macrophage responses. Macrophages along with endothelial cells play an integral part in eliciting the first stages of an immune response. Although you show a significant reduction in porcine epitope aGal, do you have the capacity to test the effect of sodium periodate on endothelial cells along with human macrophage response?
Ischemia-reperfusion injury dramatizes an important role in graft dysfunction. Given the fact that reactive oxygen species and mitochondrial dysfunction are part of the equation leading to ischemia-reperfusion injury, does sodium periodate increase or decrease the load of reactive oxygen species in your experimental conditions?
Minor: Line 279: Please complete the sentence.
All of my best regards.
Author Response
Dear authors, thank you very much for allowing me to read your work. You describe a method of quenching endothelial cell epitopes (porcine endothelial cells originating from aortic tissue) with the use of sodium periodate. Since Xenotransplantation has real-life applications your work is relevant. I have the following comments:
- We would like to thank this reviewer for the detailed analysis of our manuscript and the suggestions to improve its quality.
Sodium periodate has been known to modulate macrophage responses. Macrophages along with endothelial cells play an integral part in eliciting the first stages of an immune response. Although you show a significant reduction in porcine epitope aGal, do you have the capacity to test the effect of sodium periodate on endothelial cells along with human macrophage response?
- We fully agree that subsequent experiments aiming at the immune response after oxidation, mainly the interaction of macrophages with oxidized PECs are of high interest. We believe that such studies are beyond the scope of our present manuscript and thus will be included into a follow-up study. Actually, we have currently started to implement our principle into an organ care system perfusing whole organs. This experimental set-up will allow us to investigate immune reactions towards the oxidized graft by perfusion with human blood. The macrophage response will be included as part of the innate immune-response. In addition, reactions of different cell types to the oxidation regime itself could be tested in this set-up.
We included the aspect raised by this reviewer into our discussion:
“…..of oxidized PECs as well. A rational next step would be to study the influence of NaIO4 on other cell types and cell-to-cell interaction for example with macrophages and other immune cells. In order to improve cellular…….”
In response the other comments we included a section regarding “4.7. Side effects” which also touches the point of other cell types that might be influenced by the
Ischemia-reperfusion injury dramatizes an important role in graft dysfunction. Given the fact that reactive oxygen species and mitochondrial dysfunction are part of the equation leading to ischemia-reperfusion injury, does sodium periodate increase or decrease the load of reactive oxygen species in your experimental conditions?
- We totally agree that in case of organ transplantation, IRI and other factors such as cytokine burst due to the heart lung machine will generate a very challenging microenvironment. However, here we used a cell culture model to assess the applicability of NaIO4 for oxidation of glycan epitopes to potentially prevent hyperacute immune responses. As already stated above, further studies are needed to assess the suitability of NaIO4 application in organ care systems also considering the importance of reactive oxygen species during IRI.
We did not investigate the production of ROS in this study, but oxidative stress elicited by ROS production could be reduced by addition of buffering substances like glutathione or ascorbic acid to the oxidation solution, which should be considered for further experiments.
We included the aspect into our discussion.
…. composition of oxidation solution to be applied should be further refined, including cell-protective substances such as ascorbic acid. Another future…..
Minor: Line 279: Please complete the sentence.
- We did, a punctuation point was missing.
All of my best regards.
- We thank this reviewer for her/his positive feedback on our manuscript
The manuscript "Reduction of xenogeneic epitopes on porcine endothelial cells by periodate oxidation" by Jonas Thom et al. is a novel approach to tackling immunological responses by applying sodium periodate (NaIO4) oxidation to xenogeneic glycan antigens, one of the main obstacles to transplanting pig organs to human recipients. The authors demonstrate that controlling the time and temperature of NaIO4 reactions can effectively oxidize glycan epitopes in endothelial cells in vitro while maintaining their viability.
In the methods section, a description of how the images from the live/dead analysis were obtained and quantified needs to be included. Also, specify how the % survival shown in Figures 3A and 4A was determined.
- We thank this reviewer for critically reading and revising our manuscript.
In chapter 2.10 we describe how survival rates have been calculated and included a sentence on how the respective figures were taken.
The sections now reads:
“2.10. Quantification of live/dead assays
Quantification of living and dead cells was performed on PECs seeded in well plates by using images taken with a microscope (Axio Observer A1, Zeiss) using the exact same setting and magnification for each picture. ImageJ2 image analysis…….”
Regarding the statistical analysis, the authors do not mention what p-value they consider significant, nor what the stars in Figure 3A represent in terms of p-value. Please add this information to the manuscript.
- We added respective information to section 2.12.
“……. post-tests using GraphPad PRISM software. A measured difference was considered significant with a p-value lower than 0.05 (*), 0.01 (**), 0.001 (***) or 0.0001 (****).”
The authors demonstrate that NaIO4 can oxidize glycan epitopes on the surface of endothelial cells. Although the effects of NaIO4 are transient, could the authors further discuss this chemical treatment's potential and unintended impact on the myriad membrane receptors regulating cellular signaling and the composition of the membrane itself?
- We agree that the change in glycan structure will not only influence recognition by antibodies, but also other interactions mediated by glycans or glycosylated proteins. NaIO4 itself and by-products of the oxidation such as the created aldehydes might further influence receptors and ligands present on the cell surface. A detailed investigation of such effects will become relevant in later stages of the project.
We changed our discussion accordingly and added a new chapter "4.7. Side effects” (new references were included in the reference list).
“4.7. Side effects
In the current study, we focused our analyses on xenoantigens present on endothelial cells. However, our findings upon oxidation of tissues suggest that other cell types will be oxidized as well. Multiple different effects of NaIO4 treatment such as the inhibition of migration of macrophages or a dramatic increase in platelet aggregation have been described in other studies. (23, 24) The underlying cause could be the changes in intermolecular interaction of glycans or glycosylated molecules for example in cell metabolism, membrane integrity or receptor-ligand interactions that are all influenced by the oxidation of vicinal diols. (25) Therefore, in addition to vitality, changes to cellular behavior should be considered in future experiments.”
The discussion section lacks references when citing others' work (lines 443-445) and a more thorough discussion of the current literature. How do the results from this study compare to other reported approaches to reduce the immunogenicity of glycan antigens in xenotransplantation? Please expand the discussion.
- We added information regarding other strategies to minimize immunogenicity of xenotransplants to the discussion. However, a direct comparison of our data to the literature is limited due to the lack of similar studies. We could only find one manuscript that observed also cytotoxicity upon exposure to NaIO4.
The discussion now reads:
“……significantly improved cellular survival rates in comparison to oxidations conducted at 37°C. In previous studies high level of cytotoxicity of Madin-Darby bovine kidney (MDBK), A549 and Vero cell treated with up to 100 mM NaIO4 at 37°C for 1 hour could be observed, but not when treated with 10mM. (21) In our study 2 mM NaIO4 were already cytotoxic which might highlight that primary cells are much more sensitive to NaIO4 oxidation, compared to stable cell lines. In this study,……..”
“Other strategies to reduce immunogenicity of porcine organs involve the genetic knock-out of three known xenoantigens (αGal, Neu5Gc, SDa) and the expression of complement regulatory proteins such as CD46, CD55 or CD59. (reviewed in 16) Comparable to the NaIO4 oxidation, binding of preformed antibodies to porcine cells can be reduced by the knock-outs. (5) Because, knock-outs are not sufficient to completely abolish complement activation and deposition, additional human complement regulatory proteins are expressed. (17, 18) The reason for the remaining low levels of complement activation could be not-yet identified xenoantigens, which might be targeted by the NaIO4 treatment.”
Comments on the Quality of English Language
Line 63: remove "following"
Line 86: PEC was already abbreviated in line 74
Line 95: add "to" between prior and explantation
Line 111: define FCS
Line 115: add "at" between 5 min and 400 x g
Line 178: Blood group 0, change the zero for O
Line 239: remove "was"
Line 279: B4 is missing from the title
- Thanks for improving the wording! We changed everything accordingly.

Reviewer 2 Report
Comments and Suggestions for Authors
The manuscript "Reduction of xenogeneic epitopes on porcine endothelial cells by periodate oxidation" by Jonas Thom et al. is a novel approach to tackling immunological responses by applying sodium periodate (NaIO4) oxidation to xenogeneic glycan antigens, one of the main obstacles to transplanting pig organs to human recipients. The authors demonstrate that controlling the time and temperature of NaIO4 reactions can effectively oxidize glycan epitopes in endothelial cells in vitro while maintaining their viability.
In the methods section, a description of how the images from the live/dead analysis were obtained and quantified needs to be included. Also, specify how the % survival shown in Figures 3A and 4A was determined.
Regarding the statistical analysis, the authors do not mention what p-value they consider significant, nor what the stars in Figure 3A represent in terms of p-value. Please add this information to the manuscript.
The authors demonstrate that NaIO4 can oxidize glycan epitopes on the surface of endothelial cells. Although the effects of NaIO4 are transient, could the authors further discuss this chemical treatment's potential and unintended impact on the myriad membrane receptors regulating cellular signaling and the composition of the membrane itself?
The discussion section lacks references when citing others' work (lines 443-445) and a more thorough discussion of the current literature. How do the results from this study compare to other reported approaches to reduce the immunogenicity of glycan antigens in xenotransplantation? Please expand the discussion.
Comments on the Quality of English LanguageLine 63: remove "following"
Line 86: PEC was already abbreviated in line 74
Line 95: add "to" between prior and explantation
Line 111: define FCS
Line 115: add "at" between 5 min and 400 x g
Line 178: Blood group 0, change the zero for O
Line 239: remove "was"
Line 279: B4 is missing from the title
Author Response
The manuscript "Reduction of xenogeneic epitopes on porcine endothelial cells by periodate oxidation" by Jonas Thom et al. is a novel approach to tackling immunological responses by applying sodium periodate (NaIO4) oxidation to xenogeneic glycan antigens, one of the main obstacles to transplanting pig organs to human recipients. The authors demonstrate that controlling the time and temperature of NaIO4 reactions can effectively oxidize glycan epitopes in endothelial cells in vitro while maintaining their viability.
In the methods section, a description of how the images from the live/dead analysis were obtained and quantified needs to be included. Also, specify how the % survival shown in Figures 3A and 4A was determined.
- We thank this reviewer for critically reading and revising our manuscript.
In chapter 2.10 we describe how survival rates have been calculated and included a sentence on how the respective figures were taken.
The sections now reads:
“2.10. Quantification of live/dead assays
Quantification of living and dead cells was performed on PECs seeded in well plates by using images taken with a microscope (Axio Observer A1, Zeiss) using the exact same setting and magnification for each picture. ImageJ2 image analysis…….”
Regarding the statistical analysis, the authors do not mention what p-value they consider significant, nor what the stars in Figure 3A represent in terms of p-value. Please add this information to the manuscript.
- We added respective information to section 2.12.
“……. post-tests using GraphPad PRISM software. A measured difference was considered significant with a p-value lower than 0.05 (*), 0.01 (**), 0.001 (***) or 0.0001 (****).”
The authors demonstrate that NaIO4 can oxidize glycan epitopes on the surface of endothelial cells. Although the effects of NaIO4 are transient, could the authors further discuss this chemical treatment's potential and unintended impact on the myriad membrane receptors regulating cellular signaling and the composition of the membrane itself?
- We agree that the change in glycan structure will not only influence recognition by antibodies, but also other interactions mediated by glycans or glycosylated proteins. NaIO4 itself and by-products of the oxidation such as the created aldehydes might further influence receptors and ligands present on the cell surface. A detailed investigation of such effects will become relevant in later stages of the project.
We changed our discussion accordingly and added a new chapter "4.7. Side effects” (new references were included in the reference list).
“4.7. Side effects
In the current study, we focused our analyses on xenoantigens present on endothelial cells. However, our findings upon oxidation of tissues suggest that other cell types will be oxidized as well. Multiple different effects of NaIO4 treatment such as the inhibition of migration of macrophages or a dramatic increase in platelet aggregation have been described in other studies. (23, 24) The underlying cause could be the changes in intermolecular interaction of glycans or glycosylated molecules for example in cell metabolism, membrane integrity or receptor-ligand interactions that are all influenced by the oxidation of vicinal diols. (25) Therefore, in addition to vitality, changes to cellular behavior should be considered in future experiments.”
The discussion section lacks references when citing others' work (lines 443-445) and a more thorough discussion of the current literature. How do the results from this study compare to other reported approaches to reduce the immunogenicity of glycan antigens in xenotransplantation? Please expand the discussion.
- We added information regarding other strategies to minimize immunogenicity of xenotransplants to the discussion. However, a direct comparison of our data to the literature is limited due to the lack of similar studies. We could only find one manuscript that observed also cytotoxicity upon exposure to NaIO4.
The discussion now reads:
“……significantly improved cellular survival rates in comparison to oxidations conducted at 37°C. In previous studies high level of cytotoxicity of Madin-Darby bovine kidney (MDBK), A549 and Vero cell treated with up to 100 mM NaIO4 at 37°C for 1 hour could be observed, but not when treated with 10mM. (21) In our study 2 mM NaIO4 were already cytotoxic which might highlight that primary cells are much more sensitive to NaIO4 oxidation, compared to stable cell lines. In this study,……..”
“Other strategies to reduce immunogenicity of porcine organs involve the genetic knock-out of three known xenoantigens (αGal, Neu5Gc, SDa) and the expression of complement regulatory proteins such as CD46, CD55 or CD59. (reviewed in 16) Comparable to the NaIO4 oxidation, binding of preformed antibodies to porcine cells can be reduced by the knock-outs. (5) Because, knock-outs are not sufficient to completely abolish complement activation and deposition, additional human complement regulatory proteins are expressed. (17, 18) The reason for the remaining low levels of complement activation could be not-yet identified xenoantigens, which might be targeted by the NaIO4 treatment.”
Comments on the Quality of English Language
Line 63: remove "following"
Line 86: PEC was already abbreviated in line 74
Line 95: add "to" between prior and explantation
Line 111: define FCS
Line 115: add "at" between 5 min and 400 x g
Line 178: Blood group 0, change the zero for O
Line 239: remove "was"
Line 279: B4 is missing from the title
- Thanks for improving the wording! We changed everything accordingly.

Round 2
Reviewer 1 Report
Comments and Suggestions for Authors
Dear authors thank you very much for this improved version of your manuscript. I have no further comments.
All of my best regards.